# Determinants and Dynamic Changes of Generic Quality of Life in Human Bladder Cancer Patients

**DOI:** 10.3390/jcm10235472

**Published:** 2021-11-23

**Authors:** Yuh-Shyan Tsai, Tzu-Yi Wu, Yeong-Chin Jou, Tzong-Shin Tzai, Jung-Der Wang

**Affiliations:** 1Department of Urology, National Cheng Kung University Hospital, College of Medicine, National Cheng Kung University, Tainan 704, Taiwan; 2Department of Occupational Therapy, Asia University, Taichung 413, Taiwan; tywu820@asia.edu.tw; 3Department of Urology, Ditmanson Medical Foundation, Chiayi Christian Hospital, Chia-Yi 60002, Taiwan; 01729@cych.org.tw; 4Department of Food Nutrition and Health Biotechnology, Asia University, Taichung 413, Taiwan; 5Department of Urology, Tainan Municipal An-Nan Hospital, Tainan 709, Taiwan; tts777@gmail.com; 6Departments of Internal Medicine and Occupational and Environmental Medicine, National Cheng Kung University Hospital, Tainan 704, Taiwan; jdwang121@gmail.com; 7Department of Public Health, College of Medicine, National Cheng Kung University, Tainan 701, Taiwan

**Keywords:** quality of life, bladder cancer, dynamic changes, cystectomy

## Abstract

We measured and determined the factors associated with long-term generic quality-of-life (QOL) changes in human bladder cancer patients. We utilized the World Health Organization QOL-Brief questionnaire to assess consecutive patients’ QOL at outpatient clinics of our hospital. A mixed-effects model was constructed to investigate the determinants of QOL changes according to each domain and individual item after controlling for demographic and clinical factors, as well as the effect of radical cystectomy. We also applied a kernel smoothing method to describe the long-term dynamic changes after the first definite treatment. In total, 1185 repeated measurements were collected from 343 bladder cancer patients. The mixed-effects models demonstrated that marital status, monthly income, and comorbidity with heart disease and diabetes were significant determinants among all the study participants. Regardless of the urinary diversion type, radical cystectomy contributed to lower scores for all four domains, mainly from 4–5 years after cystectomy, which declined significantly in patients who were older than 60 years. As for non-muscle-invasive bladder cancer (NMIBC) patients with preserved bladders, tumor recurrence was a major predictor for lower scores for sexual activity in the social domain. In summary, generic QOL can be independently influenced by many factors, not only cystectomy and tumor recurrence, which should be discussed with patients before treatment.

## 1. Introduction

Bladder cancer is the ninth most frequently diagnosed malignancy in the world, and high-incidence areas include Southern Europe, North America, Northern Africa, and Western Asia [1]. There is a high incidence of bladder cancer in the blackfoot endemic area located on the southwest coast of Taiwan due to chronic arsenicism [2,3]. About 70–80% of bladder cancer cases are diagnosed as non-muscle-invasive bladder cancer (NMIBC). Most of these (about 70%) readily recur after endoscopic resection, and about 15% progress to muscle-invasive status even under regular cystoscopic surveillance, adjuvant intravesical chemotherapy, or intravesical Bacillus Calmette-Guérin (BCG) immunotherapy [4]. High-risk NMIBC has a higher probability of developing into BCG-resistant tumors and progressing to muscle-invasive bladder cancer (MIBC), which requires a radical cystetomy, or more severe conditions. In contrast, 20–30% of bladder cancer cases are initially muscle-invasive, advanced, or metastatic tumors. Currently, the standard therapy for muscle-invasive bladder cancer (MIBC) is a radical cystectomy with various types of urinary diversion or reservoir. As for those patients who are unwilling or unfit for a radical cystectomy, a bladder-sparing strategy involving the maximal transurethral resection of the bladder tumor (TURBT) or a partial cystectomy plus either adjuvant systemic chemotherapy, radiation, or both is the alternative approach [5]. Systemic chemotherapy remains the mainstream frontline therapy for metastatic diseases, since immune checkpoint inhibitors remain in their infancy [6]. Moreover, the combination of chemotherapy and immunotherapy fails to show any statistically significant success [7]. Owing to the limited success of the neoadjuvant use of such a promising immunotherapy in MIBC patients before cystectomy [8], a radical cystectomy remains burdensome to most patients and urologists and requires more investigation and surveillance [9].

Many studies have reported on the QOL of bladder cancer patients, but the majority are cross-sectional and may not be applicable for reflecting long-term effects [10,11,12,13,14,15,16,17,18,19]. In fact, some of these focus specifically on the quality of life of patients with a urinary diversion [4], and some of these are not adjusted for confounding variables [20]. Moreover, there are many unmet needs in terms of supportive care for bladder cancer patients. In addition, quality of life often differs across disease groups and cancer survivorship trajectories [21]. Thus, it is necessary to investigate the determinants and dynamic changes in the QOL of human bladder cancer patients using a generic questionnaire with a longitudinal follow-up, after ensuring the control of potential confounders and comorbidities.

To promote our understanding of and precision in dealing with bladder cancer patients, the aims of this study were to explore the factors associated with QOL changes after controlling for potential confounding comorbidities and to explore the long-term dynamic changes in QOL among bladder cancer patients who did and did not receive a cystectomy.

## 2. Materials and Methods

### 2.1. Participants

This study began after the approval of the Institutional Review Board was obtained, in 2013 (NCKUH A-ER-101-219). We collected the results of self-reported QOL questionnaires filled out by bladder cancer patients who had provided consent before an outpatient visit. We enrolled patients who were at any clinical stage or were receiving any treatment option to participate, without any exclusion criteria. The treatment strategies were based on the bladder cancer treatment guidelines of the NCKUH modified from the NCCN guidelines. The participants’ QOL was measured repeatedly using the QOL questionnaire during subsequent visits and follow-ups.

### 2.2. Measurements of QOL

The abbreviated World Health Organization Quality of Life (WHOQOL-BREF) questionnaire [22] was used for measurements of QOL and contains four domains (comprising 26 items)—physical, psychological, social, and environmental. All items are rated on a 5-point Likert scale. A higher score represents a better quality of life. The domain scores are calculated by multiplying the mean score of the items in the domain by four. Thus, the score of each domain ranges from 4 to 20. With two additional locally specific items (being respected and eating), the Taiwanese version of the WHOQOL-BREF has been validated to exhibit a good test–retest reliability (correlation coefficient > 0.75), good internal consistency reliability (Cronbach’s α > 0.91), and good construct validity in patients with several malignancies [23,24,25].

### 2.3. Procedures

The participants were instructed on how to self-complete the Taiwanese version of the WHOQOL-BREF questionnaire through tablet computers. If any questions arose about the definition of a query or item, an experienced research assistant was available to clarify the meaning of the items in a standardized way. We also collected the patients’ information, including demographic (age at time of investigation, gender, marital status, education level, and monthly family income) and clinical information (age at diagnosis, clinical or pathological stage, date and types of surgery, periods of systemic chemotherapy and radiotherapy). The former was collected via person-to-person interviews for illiterate patients. The latter was abstracted from the electronic medical records of the hospital.

### 2.4. Statistical Analysis

The participants were stratified into two categories for analysis: those with NMIBC and those with MIBC or more severe conditions. A kernel-type smoother was utilized to illustrate the dynamic changes of each item and the domain scores via the open access software, iSQOL [25,26]. The timeline of the dynamic changes (duration to date for each measurement/interview) was measured in months since the diagnosis of bladder cancer. The trends in the QOL changes were compared between those patients with and without a cystectomy. Mixed-effects models were utilized to find the determinants of generic QOL while controlling for potential confounding factors, including demographic and clinical parameters. These parameters included age at the time of the measurements (i.e., ≥70 years old, 60–69 years old, and <60 years old), gender, marital status, education level, monthly family income (i.e., more or less than USD 1750), comorbidities, disease stage, whether or not a cystectomy had been performed, time since chemotherapy (i.e., more or less than 12 months since the first chemotherapy), the history of definite or salvage radiotherapy (i.e., yes or no), and the interaction between age groups and radical cystectomy. The mixed-effects models were analyzed using the IBM SPSS 20 software.

## 3. Results

### 3.1. Demographics and Clinical Characteristics of the Participants

A total of 343 bladder cancer patients completed the WHOQOL-BREF, with a sum of 1185 repeated measurements. Fifty-two (15%) patients underwent radical cystectomy (radical cystectomy + neobladder, 19; radical cystectomy + ileal conduit, 25; radical cystectomy + ileal reservoir with the Mitrofanoff procedure, 4; radical cystectomy alone, 4). Twenty-nine patients died during follow-up. Eighty-one patients (23.6%) completed the WHOQOL-BREF once only, and these patients had a higher mortality rate and rate of heart disease compared with those who completed the WHOQOL-BREF more than once (mortality rate, 28.4% vs. 2.3%; proportion of patients with heart disease, 19.8% vs. 8.8%) (Appendix A). The mean age of all the participants at the time of the interview was 67.1 ± 11.4 years, and 238 (69.4%) of them were male. Table 1 summarizes the demographic and clinical characteristics of the participants. In total, 250 of 343 (72.9%) were married or cohabiting, and there was a higher frequency of such statuses among cystectomized patients compared with those who had not undergone a cystectomy (*p* = 0.04). There were no significant differences in terms of age, years in education, and monthly family income between patients who had and had not undergone a cystectomy (*p* > 0.05). Moreover, those who had undergone a cystectomy were less often comorbid with diabetes mellitus than those who had not (5.8% versus 19.9%, *p* = 0.01). The median length of the period from bladder cancer diagnosis to the participants’ first measurements was 14.6 months. The proportion of patients who completed more than one assessment for both groups (those who received bladder-sparing treatment versus cystectomized patients) in each period can be found in Appendix A.

### 3.2. Determinants of Generic QOL Using Mixed-Effects Models

Because only 10 (4%) of the 234 NMIBC patients received a cystectomy, we analyzed the significance of the clinical and demographic factors for the score of each item of the WHOQOL-BREF based on the 224 patients who did not receive a cystectomy. Table 2 shows the results analyzed using the mixed-effects-model method. Both being married/cohabiting and having spent longer in education were the most significant factors for the enhancement of participants’ QOL in the majority of items within the four domains. Male patients had higher scores for certain items in the social and environment domains, such as being respected, physical environment, and financial resources. Patients with a monthly family income greater than USD 1750 had higher scores in several items within the environment domain than those with a lower income did. Comorbidity either with heart disease, diabetes, or other malignancies reduced the QOL score for the physical domain’s medication item. Patients with bladder tumor recurrence had lower scores for the psychological domain’s positive feelings item and the social domain’s sexual activity item (Table 2).

In total, 52 (15%) of the 343 patients received a radical cystectomy; we subsequently analyzed the influence of the radical cystectomy on the score for each item of the WHOQOL-BREF in all study subjects, as well as the influence of the clinical and demographic factors. Table 3 shows the results analyzed using the mixed-effects-model method. As previously, both being married/cohabiting and having a higher monthly family income were the most significant factors for the enhancement of the participants’ QOL for every item in the four domains, as well as comorbidity with heart disease. Diabetes comorbidity reduced the QOL scores for the three items in the physical domain, including medication, energy and fatigue, and mobility. Patients with a muscle-invasive bladder tumor or a more severe condition had lower scores for several items in the physical, psychological, and environment domains. Patients who received a radical cystectomy scored lower for the sexual activity item in the social domain. Interestingly, such a negative effect of a radical cystectomy on sexual activity was less apparent in patients aged 60–69 years and ≥70 years compared with those aged less than 60 years (Table 3).

### 3.3. Dynamic Changes in QOL with Radical Cystectomy

#### 3.3.1. Domain Score

As for all the study subjects, the cystectomized patients in the subgroup of patients with MIBC or more severe conditions had lower QOL scores for all four domains, particularly 5 years after their initial definite treatment, compared with patients who received bladder-sparing treatment. Moreover, there was a significant decline in the score for the social domain in NIMBC patients who received a cystectomy compared with those who were treated via a bladder-sparing strategy (Figure 1).

#### 3.3.2. Item Score

In the NMIBC subgroup, cystectomized patients had significantly lower scores for six items, including safety and security, physical environment, information acquiring, and home environment in the environment domain; activities of daily living in the physical domain; and sexual activity in the social domain, compared with patients who received bladder-sparing treatment. These differences occurred mainly in the first 40 months after the first definite treatment. Patients who received a radical cystectomy exhibited persistently lower scores for the sexual activity item in the social domain than those who received bladder-sparing treatment (Figure 2A). In the subgroup of patients with MIBC or more severe conditions, cystectomized patients had significantly lower scores for three items, including body image in the psychological domain and working capacity and sleep and rest in the physical domain. These differences became prominent from 40 months after the first definite treatment (Figure 2B).

## 4. Discussion

Our study demonstrated the long-term dynamic changes in the QOL of patients with bladder cancer and their determinants. The results showed that cystectomized patients exhibited consistently lower QOL scores for each domain, starting from 40–50 months after the first definite treatment. Among them, NMIBC patients who received a cystectomy had worse QOL scores for the physical, psychological, and environment domains initially, but the differences compared with the patients with intact bladders disappeared 2 years after the first definite treatment, except for the persistently lower score for the social domain in cystectomized patients. The mixed-effects-model analysis also showed that both cystectomy and superficial bladder tumor recurrence had significant negative effects on the sexual activity item of the social domain. Most interestingly, the influence of a cystectomy on sexuality declined in patients older than 60 or 70 years, as shown in the areas of significant interaction for those aged over 60 years. These results highlight the importance of discussing private information such as sexuality before making a decision on treatment options.

This study collected long-term QOL data and comprehensively controlled for the potential confounding variables of different demographic and clinical factors through mixed-effects models. We found that being married or cohabiting, having a higher monthly family income, and having spent longer in education positively increased patients’ QOL score, which corroborated many previous reports [27,28,29,30,31]. The negative effects of an advanced stage of cancer and comorbidity with heart disease and diabetes were also consistent with previous clinical observations [9,32]. In addition, younger patients’ sexual activity was negatively affected by radical cystectomy [17,33]. In other words, these findings corroborated the validity of our models. Cystectomized patients may worry about the occasional foul odor from using diapers or urine bags and feel embarrassed while going out or participating in different social activities, which seems to have resulted in lower scores for most items in the physical and social domains, especially for the item of sexual activity. Not only can these determinants help clinicians to predict patients’ satisfaction in terms of their QOL, but they also serve as an initial platform for shared decision making and help to optimize the value of different interventions.

Bladder cancer is a solid malignancy that has high cost burdens due to its high recurrence rates, intensive surveillance strategies, and expensive treatment costs [34]. After overcoming the risk of treatment morbidity and mortality, bladder cancer patients and their family members still have the heavy burden of non-medical expenditure. Our data demonstrated that certain factors influenced the scores for some items in the environment domain, including age, marital status, family income, years in education, and comorbidity with heart disease. In fact, patients must pay at least the following additional costs: transportation for each clinic visit, caregiver-time, urine collection bags and/or diapers, and productivity lost due to illness [35]. As there are so many items that are not included in the reimbursement schedule of our National Health Insurance system, these additional costs usually bring a heavy financial burden to patients with bladder cancer and affect almost every item and domain of their QOL questionnaire. Therefore, healthcare professionals must pay attention to these issues and make early referrals to social workers for access to social welfare and/or other support.

In this study, we found that heart disease and diabetes mellitus seemed to be more common in the non-cystectomy group. This might be because patients with heart disease and diabetes mellitus might be less able to undergo major surgery (radical cystectomy) and so choose to keep their bladder. We believe that our data reflect reality and could acceptably represent the population of patients with bladder cancer.

There are five limitations to this study. First, all participants were recruited from one medical center, which might limit the study’s generalizability. However, as we included various types of patients and controlled for all demographic and clinical factors in our modeling, our conclusion regarding sexual activity would probably still be valid. Second, there may be bias between NMIBC patients and patients with MIBC or more severe conditions in terms of the number of patients who underwent a cystectomy. Therefore, we analyzed the QOL data in NMIBC patients who did not undergo a cystectomy, which may have reduced the bias in the NMIBC group. Although the number of NMIBC patients who underwent a radical cystectomy was small, NIMBC patients usually achieve long-term survival. Therefore, the long-term dynamic changes, as shown in Figure 1E–H and Figure 2A, can still provide some useful information in terms of generic QOL. Third, we did not categorize cystectomized patients into subgroups according to the accompanying diversion type, due to the limited sample size. Fourth, there were only four patients who received a radical cystectomy without any urinary diversion. All of them had ESRD and were on dialysis. Indeed, dialysis may influence QOL negatively. In fact, not only dialysis but also other factors (such as chemotherapy, radiotherapy, subsequent secondary malignancy, or other comorbidities) can greatly impact certain aspects of the generic QOL. Therefore, we used a mixed-effects model to reduce the effects from these confounders. Lastly, it seems inappropriate to make comparisons between non-metastatic MIBC patients who received a cystectomy and metastatic patients who did not receive a cystectomy owing to having unresectable tumors. However, some metastatic patients who responded well to initial systemic chemotherapy received a radical cystectomy and diversion and showed favorable outcomes because of the pathological downstaging response. Similarly, the mixed-effects-model method we used can reduce this bias after controlling for these confounders.

## 5. Conclusions

In this study, we demonstrated the long-term QOL changes in human bladder cancer patients according to a generic questionnaire. In general, marital status, monthly income, and comorbidity with heart disease and diabetes are significant determinants. Regardless of the urinary diversion type, radical cystectomy contributes to lower scores for all four domains, mainly from 4 to 5 years after the cystectomy, which declines significantly in patients older than 60 years. As for NMIBC patients with preserved bladders, tumor recurrence is a major predictor for lower scores for sexual activity in the social domain. These data reflect the general concerns of bladder cancer patients after the first definite treatment, which should be discussed with patients before deciding on the treatment selection.

## Figures and Tables

**Figure 1 jcm-10-05472-f001:**
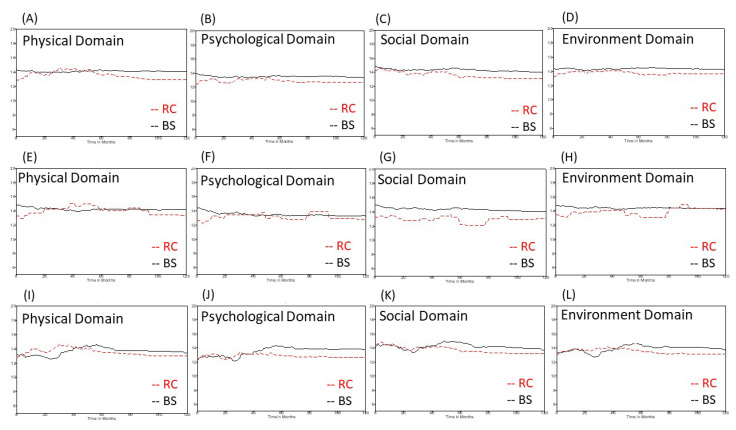
Dynamic changes in the scores for the four domains in the WHOQOL-BREF questionnaire depending on whether patients had or had not received a radical cystectomy. (**A**–**D**) all study subjects, (**E**–**H**) non-muscle-invasive bladder cancer patients, (**I**–**L**) patients with muscle-invasive bladder cancer or more severe conditions. WHOQOL-BREF, World Health Organization quality of life questionnaire brief version; RC, radical cystectomy; BS, bladder sparing.

**Figure 2 jcm-10-05472-f002:**
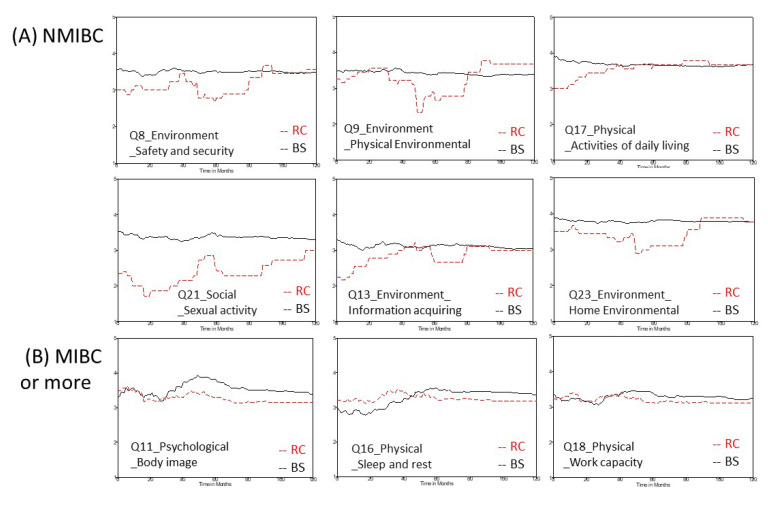
Dynamic changes in scores for certain items in the WHOQOL-BREF questionnaire according to whether or not patients received a radical cystectomy. (**A**) NIMBC patients, (**B**) patients with MIBC or more severe conditions. WHOQOL-BREF, World Health Organization quality of life questionnaire brief version; RC, radical cystectomy; BS, bladder sparing; NMIBC, non-muscle-invasive bladder cancer; MIBC, muscle-invasive bladder cancer.

**Table 1 jcm-10-05472-t001:** Demographic and clinical characteristics of bladder cancer patients.

Parameters	All	Cystectomy		*p* Value
		No	Yes	
Total no. of patients	343	291	52	
Total no. of measurements	1185	963	222	
Measurements per person	3.4	3.3	4.2	
Age (years); mean ± SD	67.0 ± 10.5	67.0 ± 11.8	67.7 ± 9.4	0.61
Age group				0.19
≥70 y/o	144	123	21	
60–69 y/o	99	79	20	
<60 y/o	100	89	11	
Gender (male/female)	238/105	192/88	46/17	0.59
Disease status (N)				<0.0001
NMIBC	234	224	10	
MIBC or more severe condition	109	67	42	
Measurements per person				Not assessed
NMIBC	3.4	3.4	3.6	
MIBC or more severe condition	3.6	3.1	4.0	
Education (years)	9.0 ± 4.7	9.0 ± 4.8	9.1 ± 4.6	0.93
Marital status (married or cohabiting/other)	250/93	218/73	32/20	0.04
Monthly family income >USD 1750 (yes/no/missing)	106/229/7	94/192/5	12/37/2	0.24
Months since first treatment (median, quartile deviation)	14.6 (30.0)	14.5 (30.3)	14.9 (27.5)	-
Comorbidity				
Diabetes mellitus	61 (17.8%)	58 (19.9%)	3 (5.8%)	0.01
Heart Disease	39 (11.4%)	37 (12.7%)	2 (3.8%)	0.06
Other malignancies	12 (3.5%)	7 (2.4%)	5 (9.6%)	0.01
Systemic chemotherapy in the past year	17 (5.0%)	13 (4.5%)	4 (7.7%)	0.32
Radiotherapy	11 (3.2%)	10 (3.4%)	1 (1.9%)	0.57

NMIBC, non-muscle-invasive bladder cancer; MIBC, muscle-invasive bladder cancer; SD, standard deviation; USD, United States dollar.

**Table 2 jcm-10-05472-t002:** Regression coefficients of the scores for each domain and item of the WHOQOL-BREF based on a mixed-effects model in NMIBC patients.

Item of WHOQOL-BREF	Age ^a^	Gender ^b^	Married or Cohabiting ^c^	Monthly Family Income ^d^	Years in Education ^e^	Heart Disease ^f^	Diabetes ^g^	Other Malignancy ^h^	Recurrence ^i^
Q1: Overall QOL			−0.18 (0.08)		0.02 (0.01)				
Q2: General health			−0.22 (0.1)				−0.22 (0.11)		
**Physical Domain**			−0.84 (0.31)		0.12 (0.03)			−1.89 (0.7)	
Q3: Physical, pain and discomfort			−0.22 (0.11)		0.03 (0.01)				
Q4: Physical, medication					0.03 (0.01)	−0.36 (0.17)	−0.4 (0.14)	−0.64 (0.32)	
Q10: Physical, energy and fatigue					0.04 (0.01)				
Q15: Physical, mobility	−0.27 (0.09)		−0.26 (0.1)		0.03 (0.01)				
Q16: Physical, sleep and rest			−0.22 (0.11)					−0.56 (0.26)	
Q17: Physical, activities of daily living			−0.18 (0.07)		0.02 (0.01)				
Q18: Physical, work capacity			−0.23 (0.09)		0.04 (0.01)				
**Psychological Domain**			−0.87 (0.34)		0.08 (0.03)				
Q5: Psychological, positive feelings			−0.22 (0.1)		0.03 (0.01)				−0.21 (0.09)
Q6: Psychological, spirituality/religion/beliefs			−0.28 (0.1)		0.02 (0.01)				
Q7: Psychological, concentration			−0.21 (0.1)		0.03 (0.01)				
Q11: Psychological, body image			−0.21 (0.09)						
Q19: Psychological, self-esteem			−0.18 (0.09)						
Q26: Psychological, negative feelings									
**Social Domain**	0.48 (0.21)	−0.74 (0.24)	−0.9 (0.25)		0.05 (0.03)				
Q20: Social, personal relationships									
Q21: Social, sexual activity		−0.36 (0.1)	−0.18 (0.07)						−0.19 (0.09)
Q22: Social, social support		−0.19 (0.06)							
Q27: Social, being respected	0.26 (0.07)	−0.16 (0.07)			0.02 (0.01)				
**Environment Domain**			−0.83 (0.24)		0.05 (0.02)				
Q8: Environment, safety and security			−0.30 (0.11)						
Q9: Environment, physical environment	0.19 (0.08)		−0.28 (0.08)						
Q12: Environment, financial resources	0.29 (0.08)	−0.3 (0.09)	−0.80 (0.22)	0.62 (0.21)	0.03 (0.01)				
Q13: Environment, information acquiring			−0.41 (0.09)		0.03 (0.01)	−0.26 (0.11)			
Q14: Environment, leisure activities				0.33 (0.09)	0.04 (0.01)				
Q23: Environment, home environment			−0.17 (0.08)	0.21 (0.07)					
Q24: Environment, health services		−0.16 (0.05)	−0.28 (0.07)	0.15 (0.06)					
Q25: Environment, transportation			−0.12 (0.06)	0.17 (0.05)	0.02 (0.01)	−0.17 (0.08)			
Q28: Environment, eating									

Values in parentheses are standard errors. ^a^ Age (>70 years versus <60 years); ^b^ gender (male versus female); ^c^ married or cohabiting (no versus yes); ^d^ monthly family income (>USD 1750 versus <USD 1750); ^e^ education (years, continuous variable); ^f^ heart disease (yes versus no); ^g^ diabetes (yes versus no); ^h^ other malignancy (yes versus no); ^i^ recurrence (yes versus no); QOL, quality of life; WHOQOL-BREF, World Health Organization quality of life brief version. Yellow background color denotes decrease and green denotes increase.

**Table 3 jcm-10-05472-t003:** Regression coefficients of the scores for each domain and item of the WHOQOL-BREF based on a mixed-effects model in patients with MIBC or more severe conditions.

Item of WHOQOL-BREF	Married or Cohabiting ^a^	Monthly Family Income ^b^	Heart Disease ^c^	Diabetes ^d^	Muscle Invasiveness ^e^	RC ^f^	RC × ≥70 Years ^g^	RC × 60–69 Years ^h^
Q1: Overal QOL		0.16 (0.07)		−0.19 (0.09)	−0.24 (0.1)			
Q2: General health			−0.3 (0.13)	−0.29 (0.1)	−0.32 (0.12)			
**Physical Domain**			−1.21 (0.41)	−0.84 (0.33)				
Q3: Physical, pain and discomfort								
Q4: Physical, medication			−0.36 (0.16)	−0.48 (0.13)				
Q10: Physical, energy and fatigue			−0.28 (0.13)	−0.24 (0.11)				
Q15: Physical, mobility	−0.19 (0.1)	0.19 (0.09)		−0.29 (0.11)	0.19 (0.1)			
Q16: Physical, sleep and rest								
Q17: Physical, activities of daily living			−0.25 (0.1)		−0.3 (0.09)			
Q18: Physical, work capacity			−0.27 (0.13)		−0.26 (0.12)			
**Psychological Domain**	−0.6 (0.3)	0.68 (0.28)			−0.91 (0.38)			
Q5: Psychological, positive feelings								
Q6: Psychological, spirituality/religion/beliefs		0.28 (0.09)			−0.3 (0.13)			
Q7: Psychological, concentration								
Q11: Psychological, body image			−0.35 (0.13)					
Q19: Psychological, self-esteem		0.16 (0.08)			−0.27 (0.11)			
Q26: Psychological, negative feelings								
**Social Domain**	−0.53 (0.24)							
Q20: Social, personal relationships								
Q21: Social, sexual activity	−0.31 (0.1)					−0.77 (0.19)	0.72 (0.27)	0.55 (0.26)
Q22: Social, social support	−0.16 (0.06)	0.12 (0.06)	−0.21 (0.09)					
Q27: Social, being respected								
**Environment Domain**	−0.45 (0.23)	0.75 (0.21)	−0.63 (0.32)					
Q8: Environment, safety and security	−0.2 (0.09)	0.19 (0.08)						
Q9: Environment, physical environment								
Q12: Environment, financial resources	−0.23 (0.1)	0.5 (0.09)						
Q13: Environment, information acquiring		0.27 (0.07)	−0.27 (0.11)		−0.3 (0.1)			
Q14: Environment, leisure activities		0.2 (0.09)			−0.55 (0.13)			
Q23: Environment, home environment	−0.17 (0.06)	0.13 (0.06)						
Q24: Environment, health services								
Q25: Environment, transportation		0.15 (0.06)	−0.18 (0.09)					
Q28: Environment, eating								

Values in parentheses are standard errors. ^a^ Married or cohabiting (yes versus no); ^b^ monthly family income (>USD 1750 versus <USD 1750); ^c^ heart disease (yes versus no); ^d^ diabetes (yes versus no); ^e^ muscle invasiveness (stage II–IV versus stage 0–I); ^f^ radical cystectomy (yes versus no); ^g^ RC (>70 years versus <60 years); ^h^ RC (>70 years versus 60–69 years); QOL, quality of life; WHOQOL-BREF, World Health Organization quality of life brief version. Yellow background color denotes decrease and green denotes increase.

## Data Availability

The clinical data, including age at diagnosis, pathological stage, date and types of surgery, periods of systemic chemotherapy and radiotherapy, were abstracted from electronic medical records of University Hospital of NCKU. The collected questionnaires were obtained from clinical cancer center of University Hospital. These data are not public and only available after obtaining the IRB consent.

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
