# Peer review of "Determinants and Dynamic Changes of Generic Quality of Life in Human Bladder Cancer Patients"

_jcm, 2021, doi:10.3390/jcm10235472_

Round 1

Reviewer 1 Report

Interesting study looking at QOL in bladder cancer patients, though there are many studies reporting on QOL questionnaires that are able to narrow down based on cystectomy and also diversion type. I understand based on your sample size you couldn't further sub-stratify. Please add more about how your study is novel and adds to the current literature.

Please review for typos and grammar.

Author Response

Point 1: Interesting study looking at QOL in bladder cancer patients, though there are many studies reporting on QOL questionnaires that are able to narrow down based on cystectomy and also diversion type. I understand based on your sample size you couldn't further sub-stratify. Please add more about how your study is novel and adds to the current literature.

Response 1: Please provide your response for Point 1. (in red)

In the section of introduction, we added one literature to support our novelty “Moreover, there were a lot of unmet needs in term of supportive care for bladder cancer. Also, quality of life often differs across disease groups and cancer survivor trajectory [21].”  (Page 2, line 61-63)

Point 2: Please review for typos and grammar.

Response 2. We did correct typos and some grammar error.

Page 3, line 109 175050,000 US dollars has been revised as 1750 US dollars.

Page 5, line 160 Recurence has been revised as Recurrence

Page 7, line 191 questionairre has been revised questionnaire

Page 8, line 209 questionairre has been revised questionnaire

page 9, line 264 NIMBC has been revised as NMIBC.

Reviewer 2 Report

1) Brief summary and overall evaluation

In this study, the authors measured and determined the factors associated with long-term generic quality of life (QOL) changes in patients with bladder cancer (BCa). There are several critical points that should be extensively revised. The reviewer would like to suggest some critiques as follows.

2) Major problems

#1

In this study, the authors mainly focused on patients with NMIBC and MIBC. Therefore, the authors should exclude patients with metastatic BCa.

#2

In the introduction part, the background of BCa was too long to understand the aim of the present study. Instead, the authors should add the reason why it is important to investigate the QOL in patients with BCa.

#3

In this study, patients who underwent radical cystectomy without urinary diversion were included. The reviewer speculated that patients who undergoing dialysis were included in the present study. Because dialysis has a strong negative impact on the QOL, the authors should exclude patients undergoing dialysis.

#4

The number of patients with NMIBC who underwent radical cystectomy was too small. Therefore, the reviewer thinks that Fig. 1 (E-H) and Fig. 2 (A) which compare the QOL between patients who underwent radical cystectomy and bladder sparing have no useful information.

2) Minor problems

#1

There are typo (Page 3, line 109; 175050,000 and page 9, line 262; NIMBC).

Author Response

(1) Brief summary and overall evaluation

In this study, the authors measured and determined the factors associated with long-term generic quality of life (QOL) changes in patients with bladder cancer (BCa). There are several critical points that should be extensively revised. The reviewer would like to suggest some critiques as follows.

 2) Major problems

#1

 Point 1: In this study, the authors mainly focused on patients with NMIBC and MIBC. Therefore, the authors should exclude patients with metastatic BCa.

Response 1

For real-world information, we enrolled all the bladder cancer patients with consent. All the patients were categorized into two groups, one is NMIBC and the other is MIBC or more (including advanced or metastatic disease). So, we correct the used term in the current study from MIBC to MIBC or more. The revised sites include Table 3 (line177), line 102, Figure 1, Figure 2, etc.

#2

 Point 2: In the introduction part, the background of BCa was too long to understand the aim of the present study. Instead, the authors should add the reason why it is important to investigate the QOL in patients with BCa.

 Response 2

In the section of introduction, we added one literature to support our novelty “Moreover, there were a lot of unmet needs in term of supportive care for bladder cancer. Also, quality of life often differs across disease groups and cancer survivor trajectory [21].”  (Page 2, line 61-63)

#3

Point 3: In this study, patients who underwent radical cystectomy without urinary diversion were included. The reviewer speculated that patients who undergoing dialysis were included in the present study. Because dialysis has a strong negative impact on the QOL, the authors should exclude patients undergoing dialysis.

Response 3

There were only 4 patients receiving radical cystectomy without any urinary diversion. All of them were ESRD on dialysis. Indeed, dialysis may influence QOL negatively. Actually, not only dialysis but also other factors (such as chemotherapy, radiotherapy, subsequent secondary malignancy, or other comorbidities) can greatly impact on the certain items of generic QOL. Therefore, we used mixed effect model to reduce the effect from these confounders.  We added these statements in the 4th limitation. (Page 10, line 270-275)

#4

Point 4: The number of patients with NMIBC who underwent radical cystectomy was too small. Therefore, the reviewer thinks that Fig. 1 (E-H) and Fig. 2 (A) which compare the QOL between patients who underwent radical cystectomy and bladder sparing have no useful information.

 Response 4

Although the number of NMIBC patients who underwent radical cystectomy was small, the NIMBC patients usually have long-term survival. Therefore, the long-term dynamic change as shown in Figure 1 (E-H) and Figure 2 (A) can still provide some useful information in term of generic QOL. We added this statement into the 2nd limitation. (Page 9, line 265-268)

2) Minor problems       #1

Point 5: There are typo (Page 3, line 109; 175050,000 and page 9, line 262; NIMBC).

 Response 5

Response 2. We did correct typos and some grammar error.

Page 3, line 109  175050,000 US dollars has been revised as 1750 US dollars.

Page 5, line 160  Recurence has been revised as Recurrence

Page 7, line 191 questionairre has been revised questionnaire

Page 8, line 209 questionairre has been revised questionnaire

page 9, line 264  NIMBC has been revised as NMIBC.